# Current Status of Bioprinting Using Polymer Hydrogels for the Production of Vascular Grafts

**DOI:** 10.3390/gels11010004

**Published:** 2024-12-26

**Authors:** Jana Matějková, Denisa Kaňoková, Roman Matějka

**Affiliations:** Department of Biomedical Technology, Faculty of Biomedical Engineering, Czech Technical University in Prague, 27201 Kladno, Czech Republic; kanokden@fbmi.cvut.cz

**Keywords:** bioprinting, vascular grafts, tissue-engineered vascular grafts, extrusion bioprinting, inkjet bioprinting, decellularized scaffold, electrospun scaffolds, bioinks

## Abstract

Cardiovascular disease is one of the leading causes of death and serious illness in Europe and worldwide. Conventional treatment—replacing the damaged blood vessel with an autologous graft—is not always affordable for the patient, so alternative approaches are being sought. One such approach is patient-specific tissue bioprinting, which allows for precise distribution of cells, material, and biochemical signals. With further developmental support, a functional replacement tissue or vessel can be created. This review provides an overview of the current state of bioprinting for vascular graft manufacturing and summarizes the hydrogels used as bioinks, the material of carriers, and the current methods of fabrication used, especially for vessels smaller than 6 mm, which are the most challenging for cardiovascular replacements. The fabrication methods are divided into several sections—self-supporting grafts based on simple 3D bioprinting and bioprinting of bioinks on scaffolds made of decellularized or nanofibrous material.

## 1. Introduction

Cardiovascular diseases (CVDs) are the primary cause of death and a significant factor in disability. According to the World Health Organization (WHO), the estimated number of deaths due to CVDs rose from approximately 12.1 million in 1990, over 18.6 million in 2019, to 20.5 million in 2021 deaths globally, which is close to a third of all recorded deaths [1]. Cardiovascular diseases continue to affect more than half a billion people worldwide. It can manifest in various ways, such as heart attacks, strokes, heart failure, and blood vessel damage [1]. The main risk factors for cardiovascular disease include high blood pressure, high cholesterol, smoking, diabetes, and obesity, among others [2]. Treatment options for cardiovascular disease encompass dietary and lifestyle modifications, pharmacological interventions, and surgical procedures.

The vascular system consists of four primary components: large blood vessels (with a diameter ranging from 10 to 30 mm), medium blood vessels (with a diameter between 6 and 10 mm), small blood vessels (with a diameter of 0.5 to 6 mm), and capillaries (with a diameter of less than 500 μm) [3]. Small-diameter blood vessel replacement has emerged as a critical focus in the field of regenerative medicine because these vessels are most commonly encountered when replacement is needed [3]. Of particular importance is the coronary artery, which typically measures less than 5 mm in diameter and is highly susceptible to atherosclerosis, leading to coronary artery disease. The ramifications of atherosclerosis on coronary vessels represent a leading cause of mortality, accounting for nearly fifty percent of all deaths in developed nations, including those in Europe and the United States. Furthermore, numerous patients who require hemodialysis rely on small-caliber vessels to facilitate long-term vascular access for venous dialysis fistulas. Additionally, small-diameter vascular grafts are critical for patients who have sustained arterial injuries exceeding 2 cm, typically resulting from traumatic events such as falls or automobile accidents [3].

When necessary, vascular surgery may use endovascular techniques such as angioplasty, stent placement, or atherectomy to widen a narrowed artery or remove a blockage (Figure 1). Alternatively, a vascular graft can replace or bypass the affected or obstructed blood vessels. Despite advancements and increasing acceptance of endovascular surgery in recent years, vascular bypass surgery remains widely performed and is considered the preferred option for patients who need long-term revascularization and have a life expectancy of over two years [4,5]. In contemporary medical practice, the use of both autologous and synthetic vessels for the replacement of small-diameter blood vessels is prevalent [6].

At present, the most commonly employed vessels for vascular grafting procedures are autologous arteries or veins. Although arteries such as the internal thoracic (a. thoracica interna) and radial arteries (a. radialis) have shown excellent patency results [7], the great saphenous vein (v. saphena magna) remains the predominant choice for autograft procedures [8]. This preference is attributed to the limited availability of arteries and the more significant complications associated with their harvesting compared to veins. The great saphenous vein is the so-called gold standard; however, it demonstrates a patency rate of only 60% over a ten-year period [9]. Furthermore, due to medical conditions, especially metabolic disorders such as diabetes mellitus, the use of autologous graft is not possible in up to 30% of the population. This is due to the lack of healthy autologous vessels and potential post-surgery complications, including low patency rates, accidental graft damage, and donor site morbidity [10].

Synthetic vascular grafts are an alternative to autologous blood vessels. They exhibit satisfactory long-term results when deployed in large-diameter arteries, such as aortoiliac replacements, where patency rates approximate 90% [11]. They are also effective in medium-diameter arteries, including replacements for the carotid or common femoral arteries. However, due to limited patency, their application is restricted to small-diameter vessels, which encompass coronary arteries, infrainguinal arteries, and infragenicular arteries [12].

The two most used materials for synthetic grafts, expanded polytetrafluoroethylene (e-PTFE) and polyethylene terephthalate (PET, Dacron^®^), are successfully used for aortoiliac replacements and in arteries with a medium diameter [13]. However, these materials have not yielded satisfactory results in small-caliber grafts due to thrombus formation or inadequate patency rates. The utilization of synthetic grafts is recommended solely when no suitable autologous vessel is present, because their failures often occur due to thrombosis, intimal hyperplasia, atherosclerosis, and infections. Therefore, the patency rate for coronary bypass grafts utilizing artificial vascular grafts is reported to be merely 32% [14]. Furthermore, it is essential for the surfaces of synthetic materials to be endothelialized or otherwise secured to mitigate the risk of thrombus formation. Then, notable improvements in patency have been achieved; however, these grafts have not surpassed the performance of autologous vessels.

If autologous veins are not available, cryopreserved cadaveric allogeneic veins can be used as an alternative to synthetic grafts. However, their use is limited by the need for cryopreservation and the decline in endothelial function due to harvesting and storage. In addition, viable allogeneic cells can induce an immune reaction, which compromises graft patency [15].

As a result, there is an urgent need for the development of laboratory-generated vascular grafts as a promising alternative to vascular surgical procedures. Current tissue engineering methods seek to create an artificial blood vessel that can grow, adapt, and heal within the body while eliminating the need for surgical autograft using a variety of methods, scaffolds, hydrogels, cell cultures, and growth factors [16]. The use of decellularized natural tissues as a framework for regenerating new veins and advancements in nanotechnology and 3D tissue engineering enables the fabrication of vein replacements with intricate microarchitecture and optimal biomechanical properties [17]. Using the bioprinting method, the carrier hydrogel with cells and growth factors is precisely positioned, and the material is of the ideal thickness to allow diffusion of nutrients throughout the volume. The incorporated cells then mature and differentiate to form living tissue that mimics the properties of natural blood vessels [18]. This innovative approach holds great promise for the development of personalized and biocompatible vascular substitutes tailored to individual patient needs [19].

Polymeric hydrogels are three-dimensional networks of crosslinked hydrophilic polymer chains that can hold up to 99% water by weight without dissolving [20]. They replicate the extracellular matrix, providing a hydrated environment that supports efficient cell seeding. Natural and synthetic materials serve as important materials for synthesizing hydrogels. Natural polymers, like gelatin, collagen, alginate, or chitosan, are typically favored in research, although they achieve worse mechanical properties than synthetic materials [20]. Conversely, when compared to synthetic polymers like plastics, elastomers, and synthetic fibers, these materials offer excellent biocompatibility and have the potential to be biodegradable. Cells can be incorporated into the hydrogel to form a cell ink that can be precisely extruded during the bioprinting process [21]. The structure of the hydrogel tube can be further supported by a scaffold, which usually has a nanospun or decellularized origin. Decellularized cardiac scaffolds derived from xenogeneic or allogenic sources offer distinct benefits, including the preservation of extracellular proteins, which in turn help maintain mechanical properties [21]. Moreover, the surface area that is accessible for cell attachment is enhanced. In addition, the incorporation of growth factors and drugs, either applied on the surface or integrated within the material, can promote both the growth and differentiation of cells [22].

Endothelial cells have the ability to organize spontaneously into tubules, a property that can also be used in the design of vascular substitutes [23]. In contrast to fully artificially formed vessels, cells are cultured with the aim of spontaneous vessel formation, and control over cell growth and distribution is quite limited. In contrast, the process of vessel fabrication using bioprinting can be relatively well controlled by the printing process parameters, but usually, additional methods are required to promote the arrangement and differentiation of cells into vascular structures. Dynamic systems with flow or pressure stresses applied to cultured cells are often used for these purposes. This article focuses exclusively on artificial vascular constructs for small-diameter vessel replacement, without the use of natural vascularization by endothelial cells, with a channel diameter of at least 1 mm.

## 2. 3D Bioprinting for Vascular Replacements

The development of artificial vascular systems presents significant challenges, particularly in creating tubular structures that are not anchored to any substrate. These structures hold promise as artificial blood vessels in three-dimensional tissue models and may have future applications as grafts for small-diameter blood vessels in clinical settings. Previously, research was concentrated on two primary methodologies: the fabrication of tubular support structures devoid of cellular material and the construction of tubular formations composed of cell−matrix combinations or cellular clusters. The origins of vascular replacement date back to the 1980s when Weinberg and Bell created an in vitro model of blood vessels, incorporating endothelial and smooth muscle cells. Its strength was derived from collagen layers and a Dacron mesh. Following that, numerous tissue-engineered vascular grafts (TEVGs) have been created and tested in animal studies; however, only a limited number have moved on to clinical trials in humans. The current methods are based on the creation of vascular substitutes using a bioprinting method that combines the use of hydrogel, cell culture, signals (growth factors, etc.), and other support materials. The three elements rely on one another and are crucial for the formation of new engineered tissue. If the tissue is made up of the individual’s own tissue, these constructs would, in theory, be resistant to thrombosis, have a lower risk of infection and possess the ability to grow [11].

An ideal vascular graft must have several critical properties to ensure its effectiveness and safety for its intended incorporation into the body’s vascular system (see the graphic in Figure 2). These laboratory-prepared grafts must meet a number of specifications to withstand blood flow and the associated pressure and strain so as not to rupture or develop an aneurysm. In addition, it is essential that they do not create any obstructions to the flow, thereby preventing the formation of large pressure gradients. Irregularities in shape could also disrupt the speed and type of flow, which could cause damage to the surrounding native vessel [24,25].

In addition, the graft surface must not induce thrombus formation, as this can lead to small vessel obstruction and other complications. The materials used for the grafts should be non-toxic and ideally biocompatible so as not to provoke unwanted immunogenic reactions. In addition, an ideal graft should have the ability to grow, remodel, and self-repair within the host organism. Next, it should exhibit reasonable compliance while maintaining fracture resistance, allowing safe handling during implantation and suitability for suturing. Finally, the manufacturing process should allow for mass production at an economically acceptable cost [26].

## 3. Hydrogels for Cardiovascular Applications

A material that has the capability to be printed with living cells, growth factors, DNA, or drugs within a delivery medium such as hydrogels, thereby enabling the creation of 3D structures with or without external stimulations, is commonly referred to as ‘bioink’ [27]. Key attributes of bioink include low viscosity, suitable biodegradability and biocompatibility, enhanced cell adhesive properties, printability, and high mechanical strength. They are important for supporting cellular and biological activities in the bioprinting process when attributes like (non-toxic) degradation, mechanical properties, and chemical structure must be controlled [28].

However, these properties limit the range of usable biomaterials for 3D bioprinting. Consequently, only a limited variety of hydrogels are currently available as bioink for 3D bioprinting. Hydrogels can also be employed as a substrate material when their viscosity is higher and/or the nozzle orifice diameter is extremely small [29]. Protein hydrogels are frequently utilized in tissue engineering because they bear resemblance to the extracellular matrix (ECM) of tissues. These polymers can either exist naturally or be created artificially [30]. The choice of bioinks for bioprinting tissue and organ substitutes is influenced by the mechanical, structural, and biological properties of the specific tissue and organ being targeted [19].

Ultimately, the crosslinking method of the hydrogel influences not only the mechanical properties but also biocompatibility and the immediate setting time post-printing. The crosslinking process leads to the formation of covalent, hydrogen, and ionic bonds and van der Waals forces between polymer chains [31]. Furthermore, the crosslinking process is employed to alter different attributes of polymer materials, including their thermal, mechanical, and physicochemical characteristics [31]. In gels that are chemically crosslinked (e.g., radical polymerization, chemical reaction of complementary groups, and enzymes), covalent bonds form between various polymer chains. This procedure yields hydrogels that have both mechanical durability and the capability to take in water and various liquids reversibly. In addition, these substances are recognized for their stability in biological systems and have an extended period of degradation [32]. Conversely, in physically crosslinked gels (e.g., ionic interaction, crystallization, and protein interaction) under the influence of temperature, pH, microwave, UV, and gamma radiation, the physical interactions present among different polymer chains inhibit the dissolution [31,33].

Recently, there has been increasing interest in physically crosslinked gels. The primary motivation for this trend is the elimination of crosslinking agents in the creation of these hydrogels. These reagents can undermine the stability of the materials being encapsulated (like proteins and cells), and they are frequently toxic substances that need to be removed from the gels before they can be used [33]. These compounds can be next harmful and toxic to humans and the environment and require extra purification steps, consuming energy, water, and solvents, thereby contributing to global warming. An alternative method is using natural crosslinking agents like genipin, citric acid, or tannic acid, which can create adequate hydrogels for bioprinting [31]. In certain situations, physical crosslinking techniques may not be enough, resulting in the active substance being released too rapidly on multiple occasions [34].

### 3.1. Natural Hydrogels

Around 90% of the polymers used in bioprinting come from natural sources [35]. Usually, natural hydrogels are water-soluble [36]. This reduces their mechanical stability, but as they are very similar in composition to the extracellular matrix, they result in an environment suitable for cell survival, which allows for further growth and division and the formation of their own extracellular matrices, which then forms the basis of future tissue [37]. Polymers in the form of hydrogels are the most commonly represented material for cardiovascular applications, especially collagen, decellularized extracellular matrix, and gelatin, less frequently fibrin and other polymers such as elastin, laminin, silk fibroin, and globular proteins such as lysozyme, BSA, and ovalbumin. For example, natural hydrogels (e.g., collagens, gelatins, and fibrinogens) are predominantly used for regenerating soft tissues such as muscles because of their similarity to the native ECM structure and composition, which fosters cell attachment, proliferation, and differentiation [38]. Biodegradable natural polymers are often used for bioprinting, as their natural remodeling into the ECM, and thus, complete integration into the cardiovascular system is exploited. These hydrogels include gelatin, collagen, alginate, chitosan, fibrin, hyaluronic acid, and their combinations, which can transition from sol to gel states in reaction to different external stimuli like temperature, light, pH, magnetism, and electricity [39].

#### 3.1.1. Collagen

The primary function of collagen is to provide mechanical support to tissues as the main extracellular matrix (ECM) protein. Collagen has a complex quaternary structure, and there are up to 29 different types of collagens, with collagen I being the most commonly used in hydrogel production [40,41]. Collagen is inherently biocompatible with low antigenicity and a low inflammatory response and is biodegradable [42]. It promotes the adhesion and growth of endothelial and smooth muscle cells, which are essential for the development of functional blood vessels, as they are the main factors in the graft’s ability to remodel and integrate into the host vascular system [19]. Collagen can be easily modified to create various forms, such as hydrogels, sponges, or films, allowing researchers to tailor the mechanical and biochemical properties of the grafts to specific clinical needs.

Limitations associated with collagen hydrogels include the potential for collagen degradation by-products to induce blood clot formation, which can result in premature graft failure [43]. In addition, collagen requires additional physical or chemical crosslinking to compensate for its lack of mechanical strength [44]. The stiffness of hydrogels is multiple times enhanced by collagen crosslinking, while energy dissipation is decreased [45]. However, there are studies that use highly concentrated collagen with a concentration of at least 10 mg/mL, which should make additional support materials or crosslinking methods unnecessary [46,47,48]. Alternatively, augmentation of collagen with other materials, such as synthetic polymers, serves to increase the robustness and durability of grafts [49]. Finally, the acquisition cost of commercial pure collagen is also very high. Thus, many research groups isolate collagen from various materials such as porcine skin, tendon, or cartilage [50].

#### 3.1.2. Decellularized Extracellular Matrix

Decellularized extracellular matrix (dECM) is another promising material for the 3D bioprinting of vascular grafts due to its ability to mimic the natural cellular environment [51]. dECM is derived from tissues that have undergone a decellularization process. This process involves the removal of cellular components while preserving the extracellular matrix, which consists of two primary components: collagen and proteoglycans, whose primary task is to provide structural and biochemical support to cells [52]. The ECM contains growth factors and cytokines, which are responsible for transmitting signals that regulate cell growth and movement, as well as increasing biocompatibility and bioactivity [53]. dECM can be obtained from various sources (bovine, ovine, murine, porcine, and simian tissues), allowing choosing matrices that closely match the desired tissue characteristics for specific applications [54]. Due to its similar composition and capacity to support tissue growth and differentiation, the utilization of the tissue-specific ECM for tissue regeneration has gained popularity, particularly in the fields of artificial grafts [55]. In addition, the porous structure of dECM facilitates cell infiltration and vascularization, which are essential for larger tissue constructs [56]. Due to the complicated preparation because of the slow gelation and the resulting low mechanical resistance, additives are often used to accelerate the solidification process, e.g., sodium alginate [57].

Studies, such as those by Zhang et al. and Potere et al., have demonstrated the potential of dECM in enhancing the biocompatibility and mechanical properties of bioprinted tissues [58,59,60]. By incorporating dECM into bioinks, researchers have been able to create vascular grafts that better support cell adhesion, proliferation, and differentiation [58]. A study by Potere et al. [59] introduced multi-layered 3D bioprinted constructs as a promising alternative for large-blood vessel replacements with the use of alginate-gelatin bioink, which was enriched with dECM of porcine aorta. The final bioink, which was combined with a two-step crosslinking process (using calcium carbonate, glucono-δ-lactone, and calcium chloride dihydrate) and seeded with mouse fibroblast cells, showed great printability and high shape fidelity. The printed construct withstands more than 35 layers without the need for any support and preserved structural stability. The overall cell viability suggested that the dECM biomaterial is suitable to support 3D cell culture.

#### 3.1.3. Gelatin

Gelatin is a polymer that is derived from the denaturation of collagen; it represents a cost-effective and minimally immunogenic mixture that is capable of undergoing a reversible sol-gel transition at temperatures below room temperature [61]. Gelatin exhibits solubility in warm water and forms physical hydrogels at reduced temperatures [61]. Consequently, below its melting point, unmodified gelatin can be employed to enhance the viscosity of bioinks utilized in extrusion-based printing [62]. Furthermore, physically gelled gelatin can provide temporary stabilization to hydrogel-based scaffolds during the printing process [62]. However, gelatin hydrogels exhibit melting and solubilization at physiological conditions, which necessitates chemical crosslinking to establish stable hydrogel scaffolds for tissue engineering applications [63]. GelMA, or gelatin methacryloyl, is a specialized variant of gelatin that emerges from the hydrolysis of collagen, a natural protein, conducted at high temperatures. This material is crafted by chemically introducing methacrylate groups to the amine-containing side chains of gelatin, a process that fundamentally alters its structure and functionality [64]. As a result, GelMA becomes a unique photocrosslinkable biopolymer, capable of undergoing crosslinking when exposed to light, which allows for precise control over its physical properties in various applications [65].

### 3.2. Synthetic Hydrogels

Synthetic polymers are typically soluble in harmful solvents or have melting points higher than the human body temperature. This poses challenges for cell viability and complicates the process of in situ cell encapsulation, leading to their representation of only about 10% of the polymers used in bioprinting [37]. However, these polymers can be readily tailored to exhibit specific properties, such as optimized mechanical characteristics and degradation rates, and can be functionalized with a wide array of bioactive agents [37]. Synthetic polymers, such as poly(lactic acid) (PLA), poly(lactic-co-glycolic acid) (PLGA), polycaprolactone (PCL), and also natural polymers, such as chitosan, calcium, and phosphate salts, are not commonly used for cardiovascular applications. They are more suitable for hard tissue substitutes due to their inorganic and osteoconductive nature, which stimulates bone growth [66]. On the other hand, these polymers are frequently used as physical frame structures for bioprinting, thereby enhancing the handling of constructs from the bioprinting stage to in vivo implantation [40,67]. Enhanced handling diminishes the risk of deformation of hydrogels or bioprinted constructs during transfer post-bioprinting, providing a stable environment for cell growth and tissue development. Although choosing tissue-specific bioinks is vital for the success of engineered tissues, the integration of angiogenic biomaterials also plays a crucial role in encouraging vascularization in these engineered tissues.

One representative commercial material is Matrigel from the company BD Biosciences (San Jose, CA, USA), which aims to create a basal matrix to promote graft survival and repair damaged tissue. It is a sterile, soluble protein extract (especially laminin, collagen IV, and enactin) derived from Engelbreth−Holm−Swarm tumors that, once prepared, forms a 3D gel that promotes cell genesis, adhesion, and differentiation [68]. Matrigel is often used as an environment for spontaneous angiogenesis, when cells differentiate into endothelium, attach, migrate and begin to form tubular structures with lumen [69]. Cocultures of endothelial cells with other cell types (stem, smooth muscle, and tumor) on Matrigel are created to investigate interactions in the tissue [69,70]. In a study by Kleinman et al. [71], Matrigel was used with other growth factors to promote neovascularization of the vasa vasorum and alleviate hypoxia to improve the remodeling of the autologous vein graft.

### 3.3. Synthetic Materials as Scaffolds for Vascular Grafts

In tissue engineering, scaffolds represent mechanical and biochemical support for cell culture, as they provide biological and chemical signals for cells. Such a carrier should resemble the extracellular matrix of native cells as much as possible; then, it supports the natural differentiation of cells growing in its structure [72]. Unfortunately, the use of scaffolds still has certain limitations, which reduce the possibility of introducing them into the clinic. So far, an ideal material has not been found that would not cause inflammation in the host’s organism; at the same time, its rate of biodegradation is synchronous with tissue or vascular formations, its degradation products would not be toxic to the host, and finally, it has similar mechanical properties to native tissue [73]. At the same time, a scaffold that faithfully imitates the native cell matrix has not yet been produced [73]. For this reason, the tendency is to produce artificial vessels without the structural support of a scaffold. However, there is potential in utilizing scaffolds for grafting small-diameter blood vessels, as we are now capable of creating models with separate tunica layers and a high degree of mimicry [74].

Synthetic polymeric conduits have become increasingly popular for bypass graft applications. Commonly used polymeric grafts include expanded polytetrafluoroethylene (e-PTFE), woven polyethylene terephthalate (PET), also known as Dacron [75,76], and polyurethanes [77]. However, due to low patency rates for small-diameter vessels (less than 6 mm in diameter), researchers have shifted their focus to exploring more compliant materials, pharmaceutical drug-loaded conduits, and tissue-engineered constructs [75,76].

Alternatively, biodegradable polymers are often used as mechanical support for blood vessels. The rate of degradation of a polymer in the body depends on both the molecular weight of the polymer and the size, or more specifically, the contact surface with the organism and the conditions in the organism [78]. Initially, there is a loss of mechanical properties, and there is a decrease in the weight-to-volume ratio. In particular, the rate of degradation is crucial for the selection of a suitable scaffold for vascular replacement [79]. In current applications, the following four synthetic, biodegradable materials and their copolymers are most often used (the biodegradation time is given in brackets):

Poly(glycolic acid) (PGA) (2–6 weeks) is a biodegradable polymer used in tissue-engineered vascular grafts (TEVGs) that degrades into glycolic acid. However, PGA alone lacks the necessary structural integrity for arterial use and can lead to aneurysms due to rapid degradation. Its mechanical properties can be enhanced through smooth muscle cell seeding or by combining it with other polymers. Additionally, PGA can be copolymerized with polylactic acid (PLA), allowing for faster degradation suitable for absorbable sutures and orthopedic implants [80].

Poly(glycerol)sebacate (PGS) (4 weeks) is a soft elastomer that promotes remodeling through rapid absorption, losing 20% of its mass in vitro within 30 days and 70% in vivo [81]. PGS grafts achieve rapid remodeling and complete endothelialization in rats, but acute obstructions lead to failures. Despite comparable insoluble elastin levels to native arteries, PGS struggles in large animal models, likely due to its rapid degradation and insufficient strength [82].

Poly(lactic acid) (PLA) (6–12 months) is also a biodegradable polymer known for its hydrophobic properties, resulting in longer degradation and early thrombogenicity [80]. Efforts to reduce thrombogenicity include cell seeding and chemical modifications, which have shown improved graft patency in rat studies [79].

Poly(ε-caprolactone) (PCL) (2–3 years) is a hydrophobic polyester with slow degradation and good mechanical properties. In rat studies, the results show a 100% graft patency rate [83]. However, significant molecular weight loss over time limits its characterization [82].

## 4. Methods of Bioprinting

The approach to creating artificial vascular replacements differs in the method of production, where the actual wall structure of the replacement, including the incorporated cells, can be printed (molded) directly, or an acellular matrix is created, and then cells and a support material (hydrogel) are applied [84]. In the classical additive bioprinting methods (extrusion, inkjet or droplet-based, and photocrosslinking method), the idea is to create a product by building it up with materials (liquid, solid, and powder) one layer at a time, although the methods can vary [38]. The next methods of vascular graft manufacturing also include molding and rolling the cell sheets into a tube, which do not directly use bioprinting and are therefore not included in this study. The primary application of an acellular matrix involves using nanofibrous or decellularized carriers, followed by the deposition of a cell layer with a bioprinter. All these approaches are illustrated in Figure 3.

Bioprinting is the process of creating bioengineered structures through additive manufacturing of biological and biologically relevant materials with computer-aided transfer and build-up processes [85]. The approach to creating artificial vascular replacements differs in the method of production. There are various solid freeform fabrication techniques, categorized into photocrosslinker-based methods, printer-based methods, and nozzle-based methods. Nozzle-based extrusion systems utilize screw- or pressure-assisted syringes to deposit continuous strands of materials, primarily for structure sizes in the centimeter range with resolutions of several hundred microns [86]. Printer-based systems, such as thermal and piezoelectric inkjet printing, produce small droplets of low-viscosity bioink using drop-on-demand systems. Their typical resolutions are in the range of approximately 85–300 µm and can be used for generating constructs ranging from millimeters to centimeters. However, the method is unable to print with highly viscous bioinks or bioinks with added reinforcements, e.g., micro- and nanofibers, due to the construction of jets. While printing-based techniques allow for combining different building materials within the assembly process, the need for supporting free-standing structures poses a significant challenge that currently limits the resolution in nozzle-based and printer-based techniques [38].

Photocrosslinker-based systems, including stereolithography and multiphoton polymerization, as well as digital light processing (DLP), utilize light for site-selective curing of photosensitive prepolymers in a bath and can achieve resolutions down to the sub-micron range [86]. On the other hand, these methods can be combined. There are products that are created on a 3D bioprinter using a nozzle-based or printer-based method, but at the same time, it is a photosensitive hydrogel that is cured using radiation of a given wavelength [87].

Another approach is methods based on molding and rolling the cell sheets into a tube, which does not directly use bioprinting and is, therefore, not included in this study. On the other hand, methods that use bioprinting on prepared materials of artificial and biological origin, such as nanofibers or decellularized scaffolds, are included in the study. Table 1 summarizes the applications of 3D bioprinting for the fabrication of vascular tubular substitutes with a diameter of at least 1 mm. These applications are discussed in more detail in the following subsections.

### 4.1. Bioprinting of Structures Using Bioinks with Living Cells

All the methods described below rely on additive manufacturing, a process that allows for the creation of 3D physical objects directly from computer-aided design data. When it comes to bioprinting vascular grafts, two primary techniques are used: extrusion-based printing and inkjet-based printing.

#### 4.1.1. Bioprinting Using Nozzle Extrusion

Bioprinting based on extrusion is one of the most widely used and, at the same time, the most affordable method of 3D printing to create 3D scaffolds because it is simple to use, flexible to operating temperatures, and inexpensive to maintain [109]. It involves the continuous application of biomaterial ink through a nozzle using a piston, air pressure, or a screw control system. Then, the structure is formed in single thin layers. However, the print resolution is relatively low (200–1000 µm) compared to with other methods [84]. It is influenced by factors such as the diameter of the nozzle, deposition speed or pressure, and the bioink’s characteristics, especially the gelation’s time and viscosity [18]. While mechanical control using a piston or screw provides precise spatial control, pneumatic extrusion is more straightforward to construct but may introduce inaccuracies in the dispensing due to delayed response of the compressed gas [110]. However, both mechanisms also allow the printing of highly viscous substances, e.g., highly concentrated collagen bioink [111]. However, extrusion of viscous bioinks from the nozzle increases shear stress, which can result in reduced cell viability in the ink [29].

One of the first developed vascular replacements using extrusion bioprinting was in the work of Smith et al. (published in 2004) [88] when a tubular structure was developed with a diameter of 2.5 mm and a height of 1.5 mm, utilizing a bioink composed of a 35% solution of Pluronic F-127 gel mixed with human fibroblasts, achieving a cell concentration of 27%. The tube was fabricated using the Bioassembly Tool (BAT, nScrypt, Orlando, FL, USA), employing either pneumatically driven or screw-controlled extrusion methods. The viability of fibroblasts following the extrusion process was approximately 60%; however, these cells were not subjected to further culture or testing. While parallel experiments were conducted to test additional layers for the prospective vessel, these components were not directly integrated with the fabricated tube [88].

Skardal et al. [90] developed a custom force transducer to measure dynamic crosslinking between two hydrogels. They used a methacrylated ethanolamide derivative of gelatin (GE-MA), which was partially photochemically crosslinked with methacrylated hyaluronic acid. Tubular constructs with a diameter of 1–2 mm were created using the Fab@Home Model 1 printer, incorporating both cell-adherent and non-cell-adherent formulations. NIH 3T3 fibroblasts were embedded in the hydrogels and printed into tube-like structures. These constructs were cultured to promote inter-gel crosslinking, fusion of layers, and tissue maturation, resulting in cell viability and the transformation of the synthetic extracellular matrix into a naturally secreted matrix. In their other research, thiol-modified gelatin and thiol-modified HA were connected using multivalent gold nanoparticles [89] or tetraacrylate derivatives of polyethylene glycol [91] in previous studies with similar objectives.

Visser et al. [92] conducted experiments using combinations of different building materials and support materials to enhance the accuracy of shaping complex structures through nozzle-based deposition methods. They created a structure resembling a vascular tree with a decreasing open vessel lumen diameter from 4 to 2 mm and a strand spacing of 0.7 mm, resulting in a 61% porosity. They obtained a 3D-printed a layered structure using PVA, PCL, GelMA-gellan (non-porous layer, containing 5 × 10^6^ chondrocytes mL^−1^), and alginate. Both concentrations were tested for cytocompatibility and the ability to dissolve the alginate component. Following 1 and 3 days of in vitro culture in a chondrocyte expansion medium, chondrocytes in all groups were found to be 75–86% viable.

Gao et al. [93] developed hollow alginate filaments with a diameter of approximately 1 mm, dependent on specific manufacturing parameters. The study involved mixing sodium alginate with mouse fibroblast cultures, resulting in filaments that had a wall thickness of 150 μm and lengths of several tens of centimeters. A custom bioprinter with a four-channel syringe pump driven by a stepper motor facilitated the fabrication process. Sodium alginate and calcium chloride solutions were dispensed through a coaxial nozzle, enabling Ca^2+^ diffusion and crosslinking to form the hollow filaments. By adjusting concentrations and flow rates, the filaments featured a gelled interior and an ungelled exterior, allowing adjacent filaments to fuse during printing, controlled by motorized XY stages. Subsequently, the researchers constructed compact structures with microchannels from these filaments, promoting nutrient diffusion to maintain cell viability, which was approximately 70% after seven days of culture.

According to MRI imaging, bifurcated tubes resembling native arteries were created from alginate, fibrin, and collagen [94]. The innovative FRESH method (freeform reversible embedding of suspended hydrogels) was used, which involves depositing a hydrogel precursor ink into a thermoreversible carrier bath. The resulting prints were highly accurate and anatomically accurate. However, no cells were incorporated into the hydrogel structure, meaning that biocompatibility and cell viability testing were not performed on these constructs. Gao et al. [95] presents a bio-blood-vessel (BBV), with a diameter of 0.5–1.5 mm, created from a study by hybrid bioink of the decellularized extracellular matrix (VdECM) and alginate using a 3D coaxial cell printing method. The BBV, which includes endothelial progenitor cells (EPC) and atorvastatin, effectively targets ischemic injury sites. The bioink supports EPC growth and differentiation, allowing for the direct creation of tubular BBVs with controlled dimensions and atorvastatin-loaded microspheres. In a nude mouse hind limb ischemia model, the BBVs showed improved EPC survival, increased neovascularization, and significant limb recovery compared to traditional cell and drug treatments.

Very promising results were produced in the work of Xu et al. [96]. A novel strategy was developed to fabricate prevascularized cell-layer blood vessels in thick tissues and small-diameter blood vessel substitutes using a four-nozzle printer with dECM and Pluronic as printing materials to print a thick structure with multilevel vascular channels having a diameter of 1–3 mm. Using this method, thick tissues using multiple levels of hollow channels are created that can be perfused before the endothelium is fully formed.

Research has also focused on the design of small-diameter blood vessels with three layers. First, the media are constructed by mixing with human aortic vascular smooth muscle cells and dECM, which changes into a gel over approximately 0.5 h. The intima is constructed by perfusing a human umbilical vein endothelial cell suspension into the inner channel of a supporting scaffold and reversing the structure by 90° every hour. The adventitia is constructed by immersing the structure in human dermal fibroblasts–neonatal suspension, which gradually attaches to the outer wall of the supporting scaffold. The entire process takes approximately 36 h [96].

Another study [87] developed a small-diameter, heterogeneous-bilayer blood vessel-like construct (a tube length of 20.0 mm, a lumen diameter of 4.0 mm, and a wall thickness of 0.8 mm) using 3D micro-extrusion bioprinting (with UV exposition and pressure-driven syringes) with gelatin methacryloyl (GelMA) bioink in different concentrations. The higher concentration was used for the inner layer containing human umbilical vein endothelial cells (HUVECs) and a lower one for the outer layer with smooth muscle cells (SMCs). The constructs were evaluated for mechanical properties, suture-ability, and cell viability. The graft demonstrated high cell viability even after 7 days of culture under static conditions [87].

The study by Bosch-Rué [97] presents a method for creating bilayered hollow fibers that mimic natural blood vessels using a triple coaxial nozzle. Alginate and collagen hydrogels are used to encapsulate human umbilical vein endothelial cells and human aortic smooth muscle cells. This process utilizes a screw-controlled extrusion printer, with the tip of the nozzle placed in contact with a 150 mM calcium chloride bath to facilitate the crosslinking of the structure. The size of the tissue-engineered vascular construct was adjusted based on printing parameters, achieving a lumen size of approximately 1.2 mm and a wall thickness of 0.1 mm. Both cell types demonstrated over 90% viability after extrusion and maintained this viability for 20 days in static culture, displaying physiological alignment within their respective layers [97].

In their other study [46], the researchers extruded a collagen structure without any support using co-axial extrusion. Although they utilized a high concentration of collagen in the bioink (10 and 20 mg/mL), which is generally considered unsuitable for cell growth and proliferation, cell viability remained relatively high 24 h after printing, at 86.1% for human arterial smooth muscle cells (HASMCs) and 85.8% for HUVECs. This experiment demonstrated the possibility of using natural bioinks without requiring radiation or crosslinking agents.

Printed constructs created by Kreimendahl et al. [98] are fabricated using the Biobots 2 3D pressure-driven extrusion bioprinter (Allevi, Cassolnovo, Italy). A fibrinogen-hyaluronic acid ink containing human umbilical vein endothelial cells (HUVECs) and human dermal fibroblasts is printed in both circular and Y-shaped designs using the FRESH printing technique. The average inner diameter of the circular constructs is 1.15 cm, while the average height of the Y-shaped constructs is 1.2 cm. The area and length of the vascular structure are three times greater compared to those of other tested bioinks [98]. Regrettably, these constructs have not been tested for longer durations, and cell viability, along with other markers, has not been evaluated.

The Kenzan method, which was developed by Koichi Nakayama [112,113], is a scaffold-free bioprinting technique that uses a specialized bioprinter to place cell spheroids precisely onto an array of microneedles, which act as a temporary support structure, according to a pre-designed 3D layout. Over time, the cell spheroids fuse together, which results in the formation of a cohesive tissue structure. After the cells have sufficiently adhered, the microneedles are taken out from the sample, leaving behind a three-dimensional tissue construct without any need for structural support. A great advantage is that the pre-designed layout and microneedles can be arranged in a way so that it is possible to create a multilayered structure using different cell types for the creation of, e.g., vascular constructs.

The study of Wang [99] creates conduits with diameters of 1 mm and 5 mm, made from a tough double-network hydrogel bioink using coaxial extrusion. This bioink combines ionically cross-linked alginate with enzyme-crosslinked gelatin, which contributes to its important mechanical properties and ability to allow fluid flow. The research demonstrated the key functions of these conduits, including the expression of specific markers and their responses to vascular stimuli.

#### 4.1.2. Inkjet Printing

Inkjet printing, also known as droplet-based bioprinting (DBB), has numerous benefits because of its simplicity, speed, flexibility, and precise control over the deposition pattern. Ink cartridges serve as bioink sources in printers, where the printhead creates droplets using thermal, electrostatic, or piezoelectric actuators [114]. It allows bioprinting with specific volumes of bioink deposition at predetermined locations, enabling the creation of spatially heterogeneous constructs with well-defined cell positioning [86]. A thermal inkjet printer utilizes a heating pulse from a thermal actuator to create a vapor bubble that expels ink droplets from the nozzle [115]. Droplet formation using the piezoelectric inkjet printing method relies on piezoelectric actuators to generate short voltage pulses that cause deformation of the bioink chamber, which helps overcome surface tension and release bioink droplets from the nozzle. By changing the applied voltage, the size and shape of the droplets can be controlled [116]. Piezoelectric inkjet printers operate at a frequency of 15–25 kHz, which can cause ultrasonic damage to cells [117]. In contrast, thermal inkjet printing heats cells for only 2 μs, resulting in minimal damage and maintaining about 90% cell viability [118]. Therefore, this method is more commonly used in tissue engineering and regenerative medicine than piezoelectric methods [119].

The fluid characteristics of the ink solution influence the size of droplets and the rate of dispensing, including factors like surface tension and viscosity, the diameter of the nozzle, and the frequency at which the printing heads are deformed [116]. For inkjet printing, it uses a restricted variety of hydrogels such as alginate, collagen, fibrin, methacrylated gelatin (GelMA), and polyethylene glycol (PEG) due to their viscosity and mechanism of crosslinking [120]. Inkjet bioprinting enables fast and accurate printing, making it ideal for creating complex patterns with high cell densities. However, the current technology is facing several obstacles despite its many benefits. These difficulties consist of a small range of bioink materials, considerable levels of cell damage caused by bioprinting, constrained mechanical and structural strength of bioprinted structures, and constraints on construct size due to insufficient vascularization and porosity [85]. The use of viscous hydrogels also affects cell viability during printing [114].

The history of droplet-based bioprinting dates back to inkjet printing technology, which originated in the 1950s [86]. Key advancements include the development of continuous-inkjet (CIJ) and drop-on-demand (DOD) inkjet printing systems in the 1960s and 1970s, respectively [84]. The concept of printing biologics was introduced in 1987, and the first inkjet-based 3D printer was developed in 2000 [121]. Subsequent advancements showed the feasibility of using modified inkjet printers to deposit living cells in a viable form, leading to the introduction of inkjet bioprinting [122]. This technology has been successfully employed for purposes such as stem cell research, tissue engineering, controlled release, transplantation, and many others.

In 2005, Kesari et al. [100] pioneered the fabrication of tubular hydrogel structures (with a diameter of 2 mm) using drop-on-demand inkjet printing. They successfully addressed the challenge of supporting the growing structure by printing liquid in liquid. This team was the first to utilize the formation of a calcium chloride (CaCl_2_)-based alginate complex in inkjet printing by printing a CaCl_2_ solution into an alginate bath using a modified Hewlett Packard printer. They then manually pipetted cell suspensions of smooth muscle cells into the hydrogel constructs before adding the next gel layer [100].

One of the first applications of DBB was the work of Zhao et al. [101] in 2012, who created a 3D vascular channel using collagen, gelatin, and HUVECs with a diameter of up to 1 mm. Gelatin functioned as a sacrificial material, with NaHCO_3_ vapor aiding polymerization. After printing collagen layers and HUVECs mixed with gelatin, the structure was kept at 4 °C to solidify the gelatin and then warmed to 37 °C for cell attachment. Finally, the gelatin was washed out, resulting in a perfused vascular channel with an average shear stress of 10 dyn/cm^2^.

In 2014, Lee et al. [102] developed a functional in vitro vascular channel featuring a perfused open lumen. The channel exclusively utilizes cells and hydrogels without incorporating any scaffolding material. The inkjet printing platform employs a robotic stage equipped with three-axis motors, an array of dispensers with microvalves, and a detachable temperature-controlling unit. The mixture of porcine skin-derived gelatin and human umbilical vein endothelial cells (HUVECs) served as the sacrificial material for the creation of fluidic channels, while a collagen hydrogel precursor (collagen concentration of 3 mg/mL; polymerized by NaHCO_3_ nebulization) was employed as the supportive matrix into which the gelatin construct was integrated. A vascular channel with dimensions of 0.7 to 1.5 mm in width and 0.5 to 1.2 mm in height was fabricated. During dynamic cultivation with a culture medium flow rate of 10 dyn/cm^2^, seeded HUVECs formed an aligned, elongated endothelial-like structure within the channel. It demonstrated the significance of dynamic culture for tissue development, while cells that were cultivated under static conditions exhibited no signs of viability.

The next works use the inkjet 3D bioprinting approach, which is based on a reverse process. A platform-assisted 3D inkjet bioprinting system has been developed to fabricate zigzag tubes with fibroblast (3T3 cells). High cell viability and high precision of the final structure are achieved, while the diameter of the droplet is about 100 µm [103]. The actual bioprinting occurs in a calcium chloride (CaCl_2_) bath, which facilitates the crosslinking of the alginate hydrogel [103]. This is advantageous, as to stack droplets or strands nearly without a stabilizing bath, it is necessary to use support materials and structures for free-form fabrication processes. These support materials have to meet various requirements, such as being removable and being able to form well-defined interfaces with building materials without repelling or blending effects [117].

In the work of Nakamura et al., who are among the first developers of DBD printers, they refined the process to produce tubular structures made from alginate. They manufactured fibers and tubes by ejecting alginate droplets into a solution of CaCl_2_. They were able to adjust the wall thickness and inner diameters of the tubular structures within the ranges of 35 to 40 µm and 30 to 200 µm, respectively, by varying the diameter of the microgel beads from 10 to 40 µm [104]. Using this method of crosslinking hydrogel in a CaCl_2_ bath, it is possible to print out tubes with lumen up to several mm [103].

#### 4.1.3. Bioprinting by Photo-Cross-Linking-Based Additive Techniques

Several methods utilize radiation crosslinking of hydrogels, differing in both the wavelength of the radiation used and the overall design of the bioprinting system. To produce tubular structures with a very high aspect ratio, traditional photocrosslinking-based additive techniques address the issue of structural support by selectively crosslinking the precursor polymer layer by layer within a reservoir of the polymer resin [123]. However, this system does not allow printing from multiple polymers, making it its biggest weakness [123].

Engelhardt [124] and Meyer [125] employed advanced additive assembly technologies to create tubular structures and bifurcated tubes using photocrosslinkable synthetic polymers and biopolymers. Their innovative approach results in the development of photopolymerizable α,ω-polytetrahydrofuranether (PTHF)-diacrylate resins, which exhibit a Young’s Modulus ranging from 8 to 28 MPa after photo-curing [125]. These resins are non-toxic and, therefore, suitable for tissue replacement. However, this method is not commonly used for vascular replacements as it has many limitations—cells cannot be incorporated into the tube material, or the dimensions of the replacements are too small to serve as replacements for small-diameter vessels. Therefore, the use of this method remains purely hypothetical at this stage. However, their tailored resins designed for inkjet-printing applications could be photo-cross-linked into carboxy-terminated polyacrylate networks, enhancing cell adhesion and proliferation [126].

The next method, most often known as laser-assisted bioprinting, uses laser pulses to precisely position cells and biomaterials on the surface. A key component of laser-assisted bioprinting (LAB) is the response of the donor layer to laser stimulation. This layer has a “ribbon” structure with an energy-absorbing layer (e.g., titanium or gold) and a bioink solution underneath. Concentrated laser pulses vaporize parts of the absorber layer, creating high-pressure bubbles that propel droplets of bioink onto the receiving substrate, where they are amplified. The original bioprinting systems using laser-based methods are known as stereolithography (SLA). SLA uses visible light or ultraviolet radiation to selectively photopolymerize layers of light-sensitive solution, creating complex cellular patterns with high submicron resolution [41].

Although the method is very accurate and, therefore, has great potential for use in bioprinting, its use is severely limited due to concerns about possible toxicity, both in terms of the light-sensitive materials used and the radiation of different wavelengths itself. Printing is also relatively complex and, therefore, expensive [127]. Several studies have reached conclusions that limit the use of LAB for cells:

Achieving an adequate cell density is challenging when compared to the natural composition [128].The necessity of using photo-curable bioink restricts the selection of available bioink options. Photo-initiators are generally toxic, which may lead to unintended cellular harm [127].The impact of continuous and prolonged laser exposure on cells has not been fully determined [129].

Therefore, the use of this method for creating vascular grafts is very rare. Anandakrishnan’s study [105] focused on endothelialized prefabricated channels (with a 1 mm lumen) in FLOAT-printed thick hydrogel models. The FLOAT method should enable rapid hydrogel stereolithography, creating centimeter-sized multiscale models in minutes. By controlling photopolymerization, a high-velocity flow of a prepolymer solution is maintained, promoting continuous growth. Initial trials showed poor adhesion of HUVECs on PEGDA models, even with fibronectin coating. By blending GelMA with PEGDA, improved endothelial adhesion was achieved with a higher GelMA content. This fast-printing method should significantly reduce part deformation and cellular injury compared to traditional 3D printing, marking a step toward large-scale engineered tissue models [105].

### 4.2. Cellularization of the Acellular Graft

#### 4.2.1. Electrospun Grafts

Although 3D bioprinting is an excellent tool for creating complex structures and layers with defined structures, there are certain limitations, e.g., mechanical strength, pore size, and convection of nutrients. Therefore, other materials that provide mechanical support and contribute to developing blood vessels with their other properties are also used [11]. One option is to create a blend of electrospun fibers and bioprinted hydrogels. Merging nanofibers with 3D bioprinting results in materials that possess adequate load-bearing strength, intricate shapes, and targeted coating characteristics [22]. Additionally, the active surface area available for cell adhesion is improved, and furthermore, the growth and differentiation of cells can be encouraged through the inclusion of growth factors and drugs either on the surface or embedded within the material [22]. Recent advances have even integrated cell-laden bioinks into the 3D bioprinting process, allowing for the direct incorporation of living cells into the scaffold. This hybrid method leverages the fine fibrous network of electrospun materials with the precise geometrical control of 3D printing, creating scaffolds that closely mimic the natural architecture of human tissues. The resulting constructs show improved cellular behavior, including better attachment, migration, and differentiation, which are critical for tissue engineering applications [130].

Depending on the particular application, 3D-printed scaffolds and electrospun filaments can be utilized in various combinations, such as electrospinning onto 3D-printed scaffolds, 3D printing onto electrospun filaments, alternating between 3D printing and electrospinning, using electrospun filaments as inks for 3D printing, embellishing or inflating 3D-printed scaffolds with segments of electrospun nanofibers, or creating electrospun scaffolds on 3D-printed templates or collectors [131]. The process of electrospinning addresses the challenge of structural support by depositing polymer fibers layer by layer onto a collector, e.g., rotating mandrel, forming tubular structures with high aspect ratios [22,132,133]. Electrospinning facilitates the alignment and stretching of polymer chains, producing nanofibers with a precisely defined structure resembling the natural extracellular matrix (ECM) characteristics. These nanofibers offer multiple sites for cell attachment and proliferation, impacting cell shape and behavior [134]. Moreover, nanofibrous structures can be used as platforms for the delivery and controlled release of drugs, peptides, or bioactive factors. However, in terms of mechanical properties, even nanofibers have their limitations [135].

One of the primary advantages of electrospun grafts is their ability to be customized using various biocompatible polymers, such as collagen [130], poly(d, l-lactide-co-glycolide) (PLGA) [136], polyethyleneoxide (PEO) [137], poly(ε-caprolactone) (PCL) [130], and poly(ethylene terephthalate) (PET). However, a notable limitation is the difficulty in combining multiple types of polymers within a single graft, which can restrict the graft’s functional versatility [138].

Electrospun grafts are being reinforced with 3D printing techniques, as shown by researchers Zhang et al. and Adhikari et al. These advancements have resulted in grafts with better mechanical properties and less kink formation, which are crucial for maintaining function under physiological conditions [132,138,139]. For example, incorporating bioactive molecules like vascular endothelial growth factor (VEGF) and vascular endothelial cadherin (VE-cadherin) into the electrospun fibers significantly improves endothelialization and integration with hosting tissues [138]. Using bioprinting on nanofiber supports was possible to produce vascular grafts with complex 3D microstructure, high porosity, and interconnection between bioink fibers. This increases both graft strength and cell adhesion and adhesion, which is essential for long-term implantation of vascular replacements [132].

Carabba and colleagues have developed a graft to closely replicate the natural structure of human arteries, which contains an internal conduit of co-electrospun gelatin and polycaprolactone nanofibers with endothelial cells deposited onto the luminal surface using a rotative bioreactor and a bioprinting system creating two cell-laden (vascular smooth muscle cells and pericytes) hydrogel layers to form the tunica media and adventitia [106]. After 6 days of maturation, graft implantation in the left pulmonary artery of swine showed excellent hemostatic properties and the ability to withstand blood pressure and remained patent at 5 weeks post-implantation.

Zhang et al.’s recent research [107] developed a 3D bioprinting system that utilizes a six-degree-of-freedom robotic arm to print cells on complex-shaped vascular scaffolds from multiple angles using extrusion from a single-channel Multipette. Artificial blood vessels with newly formed vascular branches and capillaries were prepared on a pre-constructed scaffold and a repeated print-and-culture approach. First, the tubular scaffolds made of polyl-lactic acid (PLLA) that are utilized in cell bioprinting experiments are created using the electrospinning technique. Then, an endothelial bioink is made, which involves suspending hCMEC/d3-eGFP cells in a cold endothelial medium (RPMI-1640 medium enriched with 10% FBS, 1% Matrigel, 1× GlutaMax, 1× NEAA, and 10 mM HEPES), and finally, the printing of this bioink is utilized onto a tubular scaffold using print-and-culture cycles, where material containing cells is printed and subsequently cultured through repeated printing and culturing techniques.

A biomaterial-free printing method was employed, utilizing an oil bath to aid the attachment of cells to various surfaces. In the mineral oil bath, the driving force provided by the hydrophobic oil, coupled with the prior coating of ECM protein on the scaffold, facilitates the adherence of printed aqueous bioink droplets (which include cells) to the scaffold. After printing, during the culture phase, the mineral oil in the tank is replaced with endothelial medium, and the tubular scaffold undergoes perfusion with a self-constructed bioreactor. Human endothelial cells create a robust vascular network with comparable branch point quantities after 24 h of culture on Matrigel, indicating that the angiogenic capacity of endothelial cells remains intact following printing. The printed hCMs seeded on the tubular scaffold begins to beat for a minimum of 10 days rhythmically. This bioprinting system has the ability to produce cardiovascular and contractile tissues that can survive for an extended period [107].

A new technique for producing bionic small-diameter vascular grafts was established by Jin’s team, utilizing nanofiber electrospinning in conjunction with a uniquely designed rotary bioprinter [108]. They created artificial vessel grafts composed of an electrospun PCL scaffold, a layer bioprinted with GelMA-based bioink that incorporates smooth muscle cells (SMCs) along with perfused human umbilical vein endothelial cells (HUVECs). The bioprinting approach employs a rotary pressure-driven extrusion method.

#### 4.2.2. Decellularized Grafts

Decellularized grafts present another approach for use in vascular tissue engineering. These scaffolds, which are derived from natural tissues by removing cellular components and leaving only the microstructure extracellular matrix (ECM), provide a natural and biocompatible environment that can be used for recellularization [140]. The whole process, imaged in Figure 4, involves several steps, starting with the harvesting of the tissues either from a donor or a cadaver. After that, cells are removed from the tissue using a combination of physical, chemical, or enzymatic treatments to ensure that the ECM scaffolds maintain their structural and chemical integrity [56,141,142,143]. In the end, the tissues need to be thoroughly washed and sterilized to remove any cellular debris, thus preventing any potential immune response [56,78,141]. The decellularized material prepared in this way is ready for additional techniques, such as cell seeding through bioprinting, modifying its physical and chemical properties, and incorporating biochemical signals into the fiber structure. Integrating decellularized scaffolds with various substances, such as biomaterials, pharmaceuticals, and growth factors, could enhance the properties of the decellularized scaffolds [144]. After these steps, the material can be embedded in an in vivo environment. The whole process is imaged in Figure 4. The resulting decellularized scaffold maintains natural microenvironment features such as growth factors and proteins, which are crucial for cell attachment, proliferation, and differentiation [145].

Cellularization of these acellular (decellularized) grafts involves seeding them with cells in order to restore their full biological function [140,146]. These cells can be bioprinted directly onto the decellularized scaffolds, where they integrate and form new functional tissue. Bioprinting allows for precise placement of the cell-laden bioink or cell suspension, enabling the creation of vascular structures with exact geometries and tissue organization [123,147]. This not only restores the biological function of the grafts but also ensures better integration with host tissue upon implantation [148]. The combination of 3D bioprinting with decellularized scaffolds is a promising application since the ECM from decellularized tissues supports cell attachment and growth, while 3D bioprinting technology provides precise control over tissue architecture. However, there are still challenges to address, such as degradation of the grafts [149] and achievement of adequate vascularization within the grafts to overcome poor diffusion of nutrients and gases [147].

Currently, no group is involved in the development of vascular replacements based on decellularized tissue and bioprinting, where a tubular structure would be developed to serve as a replacement vessel. Some groups only focus on the development of planar vascular patches intended to heal the disrupted vessel wall [150].

## 5. Discussion and Conclusions

Cardiovascular diseases are the most common cause of serious complications or deaths in the world [1], and therefore, the demand for cardiovascular substitutes is essential and current. Given the limited number of donors, problems with the implantation of allogeneic grafts, and sometimes problematic use of autologous grafts, the creation of tailor-made substitutes by bioprinting hydrogels with cells is a logical step for the creation of a universal substitute tailored to each patient.

This work demonstrates various methods of preparing tissue substitutes for small-diameter vessels, which effectively solve the high rate of occlusion of small-caliber vascular grafts. All available bioprinting techniques have been used to produce vascular grafts, but each has certain limitations that make it difficult to produce a mechanically robust, biocompatible structure. The use of resins does not allow for the combination of different materials [123]. Nozzle extrusion has limits in resolution. Moreover, the structures are not very mechanically strong, so they often need additional material to support them. Inkjet printing has better resolution but is still not very mechanically robust [38]. Finding the optimal manufacturing process that would combine a sufficient number of materials, be biocompatible and be sufficiently mechanically robust is a relatively difficult task.

One of the main challenges in developing artificial tissue substitutes in vitro is the provision of nutrients and the removal of waste products from deeper layers of the structure. The maximum depth at which effective exchange occurs only by diffusion is about 2 mm [151]. In thicker structures, a conduction system is required through which a liquid, usually a culture medium, flows and ensures the exchange of substances even in deeper structures, similar to the functioning of the vascular system in the organism. In the absence of such a network, necrosis occurs from the inside of the structure, which prevents the overall growth of cell mass and tissue and ultimately leads to the death of the entire culture or to the weakening of the structure of part of the substitute and its failure [151].

Furthermore, if we find a suitable bioprinting process and material, despite the successful validation of the prosthesis in vitro, implantation in vivo is necessary [152]. Despite the many technologies that are available for production today, there are still a number of shortcomings that must be resolved before implantation into a patient can occur. For example, there is no perfect simulation that verifies the material and biomechanical properties of the prosthesis to guarantee success in implantation into the host and in integration and remodeling into its vascular network. Moreover, the choice of prosthesis depends not only on the results of the clinical part, but also on economic aspects [153]. Moreover, the path from the laboratory to the market is a long one, lined with many regulations, laws, and certifications, which prolongs practical use but is essential for patient and environmental safety.

A temporary obstacle may be the access of hospitals, as well as surgeons and other medical professionals, to new technologies, especially vessels produced in laboratory conditions for clinical use. Current approaches to surgical treatment of cardiovascular diseases have been used for decades, although many changes and improvements have also been made in this area. However, artificially created vessels are a completely new technology, and it may take several years before there are enough clinical studies to support the adoption of new standards [154].

The development of tissue substitutes using bioprinting methods will continue, and the results of this research will be further improved. This can also be contributed to by the current boom in artificial intelligence, which can speed up and simplify many processes and simulations, which can also contribute to accelerating the development of new bioprinting methods [154]. In addition, the 4D bioprinting method is currently on the rise, which is due to the properties of new “smart” materials that change their properties after printing under the influence of external physical forces, such as heat and radiation. These materials can interact even after implantation directly in the organism’s environment [155].

Nevertheless, all previously discussed uses of bioprinting for synthetic vascular grafts are currently in the developmental phase and undergoing in vitro evaluation. For a 3D-printed vascular graft to receive approval for patient implantation, it must adhere to the same protocols as medical devices and products, regardless of the production method [156]. The specific type of authorization will depend on the vascular prosthesis’s risk classification [156].

A large number of research groups around the world are involved in the development of new solutions, from testing the basic properties of hydrogels and bioinks through the creation of vessels in vitro, even in vivo applications, to the final stage of clinical testing. Many of them have a high potential for success in developing a vessel that would help many patients around the planet and thus improve not only the quality of life of people but also the entire society. Although not only a lot of effort and financial resources are devoted to this process, it will also have a huge impact globally, which will have a positive effect on all areas.

## Figures and Tables

**Figure 1 gels-11-00004-f001:**
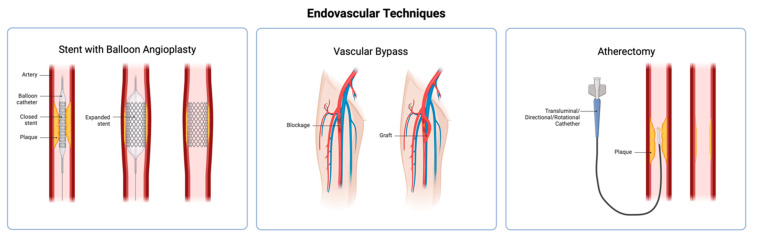
Vascular surgery techniques as an alternative for revascularization treatment. An illustration was created with https://www.biorender.com/, accessed on 12 November 2024. BioRender.com.

**Figure 2 gels-11-00004-f002:**
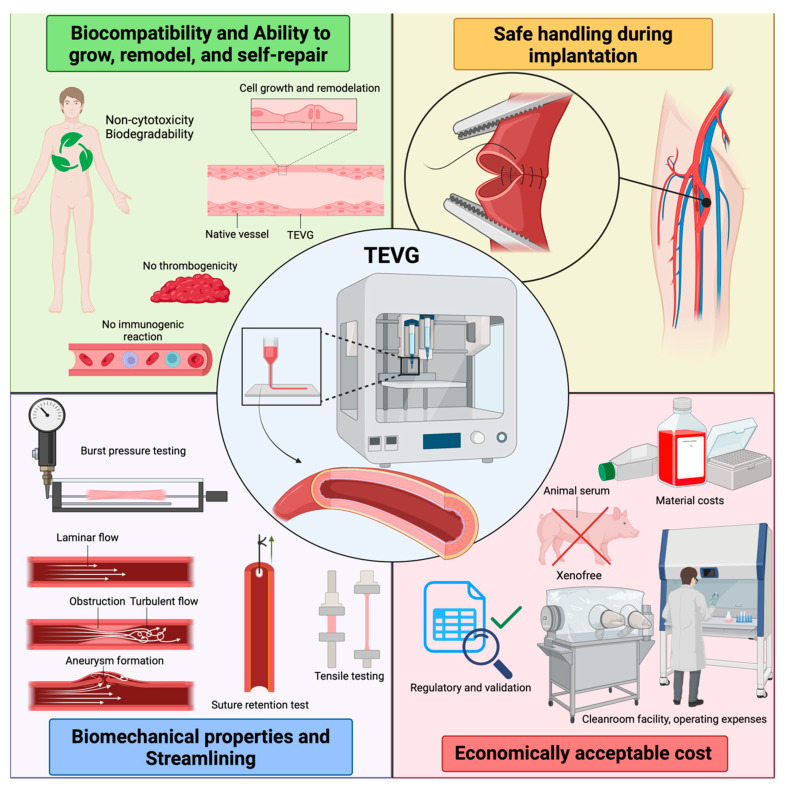
The requirements for ideal vascular replacement are broad and cover many areas. An illustration was created with https://www.biorender.com/, accessed on 28 October 2024.

**Figure 3 gels-11-00004-f003:**
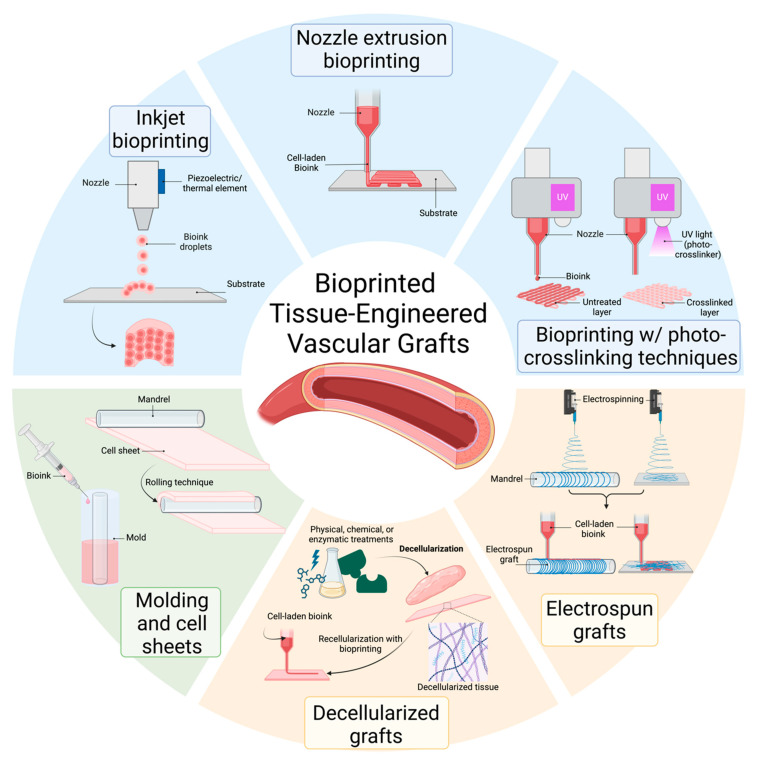
Different methods of creating vascular grafts. Highlighted in blue are classical additive bioprinting techniques, which use hydrogel, cells, and other support materials to create a custom 3D construct. The yellow-colored are replacements created by a combination of bioprinting and acellular support of decellularized or electrospun fibers. The green-colored ones are methods that involve molding hydrogels or rolling sheets with cells into tubes. Due to the absence of bioprinting, these two techniques are not the focus of the paper. An illustration was created with https://www.biorender.com/, accessed on 10 October 2024.

**Figure 4 gels-11-00004-f004:**
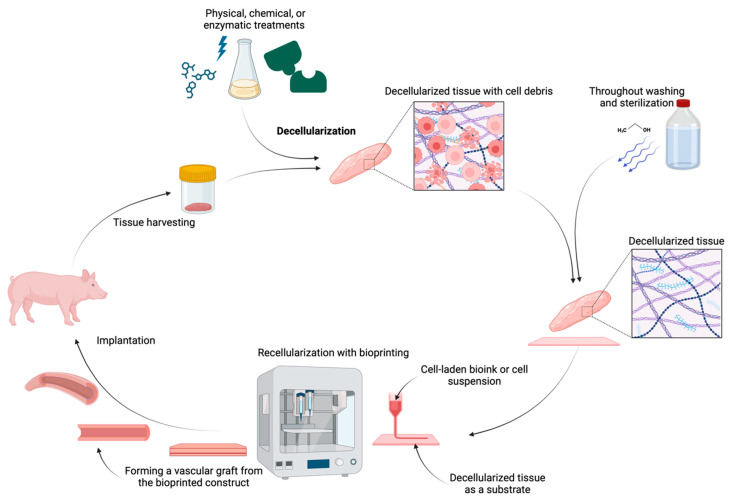
The process of bioprinting on decellularized grafts involves the following: harvesting tissues from donors or cadavers; removing cells while preserving the extracellular matrix; washing and sterilizing to eliminate debris and prevent immune responses; enhancing scaffolds with biomaterials and growth factors; preparing for techniques like cell seeding and modification; embedding in an in vivo environment to support cell attachment and growth. An illustration was created with https://www.biorender.com/, accessed on 15 October 2024.

**Table 1 gels-11-00004-t001:** Applications of 3D bioprinting.

3D Printing Strategies	Material	Dimensions of Printed Constructs	Brief Description	Reference
Extrusion methods	35% solution of pluronic f-127 gel	A diameter of 2.5 mm and a height of 1.5 mm	One of the first developed vascular replacements	[88]
Extrusion methods	GelMA	A diameter of 1–2 mm	Tubular-like structures with fibroblasts, modified also by gold nanoparticles or tetraacrylate derivatives of polyethylene glycol	[89,90,91]
Extrusion methods	GelMA-gellan hydrogel and alginate (sacrificial material)	A vascular tree with a lumen diameter from 4 to 2 mm and a strand spacing of 0.7 mm	Combinations of different building materials and support materials to enhance the accuracy of shaping complex structures	[92]
Extrusion using coaxial nozzle	Alginate	A diameter of 1 mm, wall thickness of 150 μm and lengths of several tens of centimeters	Sodium alginate and calcium chloride solutions were dispensed through a coaxial nozzle, enabling Ca^2+^ diffusion and crosslinking to form the hollow filaments.	[93]
FRESH method (extrusion-based)	Alginate, fibrin, and collagen	A diameter of 1 mm	Depositing a hydrogel precursor ink into a thermoreversible carrier bath	[94]
Extrusion using coaxial nozzle	dECM and alginate	A diameter of 0.5–1.5 mm	Fabricated bio-blood-vessels with improved endothelial cell survival, increased neovascularization, and significant limb recovery in in vivo model	[95]
Extrusion using 4 nozzles	dECM and Pluronic	A diameter 1–3 mm	Multiple levels of hollow channels made like prevascularized cell-layer blood vessels and small-diameter blood vessels with three layers	[96]
Extrusion with pressure-driven syringes and UV exposision	GelMA	A length of 20.0 mm, a lumen diameter of 4.0 mm, and a wall thickness of 0.8 mm	Heterogeneous bilayer blood vessel-like construct, evaluated for mechanical properties, suture-ability, and cell viability	[87]
Extrusion using a triple coaxial nozzle and a calcium chloride bath	Alginate and collagen	A lumen size of approximately 1.2 mm and a wall thickness of 0.1 mm	Bilayered hollow fibers that mimic natural blood vessels with 90% viability after extrusion and maintained this viability for 20 days in static culture	[97]
Extrusion using a coaxial nozzle	Collagen with high concentration (10 and 20 mg/mL)	A diameter of 1 mm	High cell viability of 86.1%/85.8% after 24 h; the possibility of using natural bioinks without requiring radiation or crosslinking agents	[46]
FRESH method (extrusion-based)	Fibrinogen-hyaluronic acid ink	An inner diameter of the circular constructs was 1.15 cm, and the average height of the Y-shaped constructs was 1.2 cm.	The area and length of the vascular structure were three times greater compared to those of other tested bioinks. Not been tested for longer durations and cell viability.	[98]
Extrusion using a coaxial nozzle	Alginate and gelatin	Diameters of 1 mm and 5 mm	The research demonstrated the key functions of fabricated conduits, including the expression of specific markers and their responses to vascular stimuli	[99]
Inkjet with a calcium chloride bath	Alginate	A diameter of 2 mm	First successful use of an inkjet printer for vascular grafts by printing liquid in liquid	[100]
Inkjet printing with NaHCO_3_ vapor aiding polymerization	Collagen and gelatin (as sacrificial material)	A diameter up to 1 mm	Vascular channels made by collagen, printed on gelatin structure, which was finally washed out	[101]
Inkjet printing with NaHCO_3_ vapor aiding polymerization	Collagen and gelatin (as sacrificial material)	A channel with dimensions of 0.7 to 1.5 mm in width and 0.5 to 1.2 mm in height	The channel utilized cells and hydrogels without incorporating any scaffolding material, developing a layer of endothelium; it demonstrated the significance of dynamic cultivation.	[102]
Inkjet printing with a calcium chloride bath	Alginate	A diameter of 1 mm	Vascular zigzag channels using an advantageous printing method in the calcium chloride bath	[103]
Inkjet printing with a calcium chloride bath	Alginate	Fibers and tubes with the wall thickness and inner diameters of the tubular structures within the ranges of 35 to 40 µm and 30 to 200 µm, respectively	Fibers and tubes made by method of reverse bioprinting in the bath	[104]
Stereolithography—FLOAT method	GelMA and PEGDA	A lumen of 1 mm	The method enables rapid hydrogel stereolithography, creating centimeter-sized multiscale models in minutes.	[105]
Extrusion on electrospun scaffolds	Alginate and Pluronic, Electrospun gelatin and PCL scaffold	A lumen of 5 mm and a length of 50 mm	The graft closely replicates the natural structure of human arteries, with excellent hemostatic properties and the ability to withstand blood pressure, remaining patent at 5 weeks post-implantation.	[106]
Extrusion printing on an electrospun scaffold using a robotic arm	Matrigel and PLLA scaffold	A diameter up to 1 mm	Artificial blood vessels with newly formed vascular branches and capillaries	[107]
Extrusion printing on an electrospun scaffold	GelMA and PCL scaffold	A diameter up to 6 mm	The rotary bioreactor for proper application of bioink and next dynamic cultivation	[108]

## Data Availability

No new data were created or analyzed in this study. Data sharing is not applicable to this article.

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
