# Peer review of "Current Status of Bioprinting Using Polymer Hydrogels for the Production of Vascular Grafts"

_gels, 2024, doi:10.3390/gels11010004_

Round 1

Reviewer 1 Report

Comments and Suggestions for Authors

Reviewer’s comments and suggestions to Authors

The most recent developments in artificial vascular constructions for small-diameter artery replacement are superbly reviewed in this publication. Since it enables precise distribution of cells, material, and biochemical signals, patient-specific tissue bioprinting is one of the alternative strategies being considered to replace damaged blood vessels with an autologous transplant, according to the report.

As seen by the large number of the papers cited in this review, the scientific community is making significant efforts to produce blood vessels with small diameters. Therefore, the informational collection, organized into parts for hydrogels and bioprinting techniques, is pertinent to the area, especially to the special issue Application of Hydrogels in 3D Bioprinting for Tissue Engineering. A table that lists the material, bioprinting technique, accomplishment, and references for each work cited in this review would enhance this manuscript because it is still difficult to optimize manufacturing processes for appropriate materials, adequate mechanical properties, and proper biocompatibility.

The accomplishments outlined in the evaluation were counterbalanced by the challenges and unresolved problems covered in the final section. Considering the issues that need to be addressed and the milestones that need to be reached, they urge the reader to introduce or delve deeper into the development of artificial vessels utilizing bioprinting techniques.

Just a few comments referring to text edition:

Line 120 Section number is 2. while the title is 3D bioprinting for vascula replacements.

Line 520 It would be useful to write the complete name of HASMCs with the acronymous in brackets. 

Line 660 Please, include in brackets the acronymous for laser-assisted bioprinting (LAB).

Author Response

Reviewer’s comments and suggestions to Authors

The most recent developments in artificial vascular constructions for small-diameter artery replacement are superbly reviewed in this publication. Since it enables precise distribution of cells, material, and biochemical signals, patient-specific tissue bioprinting is one of the alternative strategies being considered to replace damaged blood vessels with an autologous transplant, according to the report.

As seen by the large number of the papers cited in this review, the scientific community is making significant efforts to produce blood vessels with small diameters. Therefore, the informational collection, organized into parts for hydrogels and bioprinting techniques, is pertinent to the area, especially to the special issue Application of Hydrogels in 3D Bioprinting for Tissue Engineering. A table that lists the material, bioprinting technique, accomplishment, and references for each work cited in this review would enhance this manuscript because it is still difficult to optimize manufacturing processes for appropriate materials, adequate mechanical properties, and proper biocompatibility.

The accomplishments outlined in the evaluation were counterbalanced by the challenges and unresolved problems covered in the final section. Considering the issues that need to be addressed and the milestones that need to be reached, they urge the reader to introduce or delve deeper into the development of artificial vessels utilizing bioprinting techniques.

Just a few comments referring to text edition:

Line 120 Section number is 2. while the title is 3D bioprinting for vascula replacements.

Line 520 It would be useful to write the complete name of HASMCs with the acronymous in brackets. 

Line 660 Please, include in brackets the acronymous for laser-assisted bioprinting (LAB).

Thank you very much for your positive review of our article and for checking it out. We have corrected mentioned comments in the article.

Reviewer 2 Report

Comments and Suggestions for Authors

In the article titled “Current status of bioprinting using polymer hydrogels for the productions of vascular grafts”, The review provides an overview of the current state of bioprinting for vascular graft manufacturing and summarizes the hydrogels used as bio-inks, the material of carriers, and the current methods of fabrication used, especially for vessels smaller than 6 mm, which are the most challenging for cardiovascular replacements. It is worth affirming that the author has guiding significance for the review of the Current status of bioprinting using polymer hydrogels for the production of vascular grafts, and puts forward the following suggestions for the article.

1. In the introduction, the author should strengthen the description and summary of the bioprinting preparation of polymer hydrogels. The application of polymer hydrogel bioprinting preparation in vascular transplantation and its influence on vascular transplantation are emphasized.

2. After introducing the first part of the article, part 2.3, please check whether the content is accurate and comprehensive.

3. Is there only 3D printing in bioprinting technology? If there are other technologies, please summarize.

4. Quoted more outdated references, please refer to more novel references.

5. It is mentioned in the article that different methods can be used to improve the performance of the prepared samples, but the article does not reflect how the performance changes. For example, lines 209-214 mention that physical or chemical crosslinking can be used to compensate for the lack of mechanical strength, and can increase the robustness and resistance of the graft. There is no exact data to reflect the performance changes, please compare the performance data.

6. Are there sufficient discussions on the future development of polymer hydrogel bioprinting to prepare vascular grafts in the literature? Or does it rely on speculative statements that are not supported by sufficient research?

Author Response

Comments and Suggestions for Authors

In the article titled “Current status of bioprinting using polymer hydrogels for the productions of vascular grafts”, The review provides an overview of the current state of bioprinting for vascular graft manufacturing and summarizes the hydrogels used as bio-inks, the material of carriers, and the current methods of fabrication used, especially for vessels smaller than 6 mm, which are the most challenging for cardiovascular replacements. It is worth affirming that the author has guiding significance for the review of the Current status of bioprinting using polymer hydrogels for the production of vascular grafts, and puts forward the following suggestions for the article.

Thank you very much for the positive review of our article and valuable advice. We have incorporated all your comments in the manuscript and we will send your comments on.

  1. In the introduction, the author should strengthen the description and summary of the bioprinting preparation of polymer hydrogels. The application of polymer hydrogel bioprinting preparation in vascular transplantation and its influence on vascular transplantation are emphasized.
  • The paragraph about the use of hydrogels in bioprinting was added to the introduction.

  1. After introducing the first part of the article, part 2.3, please check whether the content is accurate and comprehensive.
  • Chapter 2 is a brief introduction to 3D bioprinting, as we use the terms in the article, but on the other hand, the issue is so extensive and there are already many articles describing the methods that we did not want to discuss these technologies further, as they are not the subject of our summary.

  1. Is there only 3D printing in bioprinting technology? If there are other technologies, please summarize.
  • The most commonly considered 3D bioprinting technologies in research are extrusion, inkjet, laser-assisted and stereolithography. We also mention in the article the cell plate rolling and molding methods, which however do not use precision deposition using a printer, so we do not discuss them further as mentioned here. In addition, planar biomaterial coating where a printer is used could be considered as a bioprinting method (e.g. recolonization of planar synthetic scaffolds, decellularized tissues etc.). In these cases, the bioprinter (using single layer Z level) is used as a precise coating instrument for the planar scaffold (sometimes mentioned as “2.5D”). However, in this review, we focused on volumetric scaffolds utilizing bioinks where cells are spread in 3D volume during the processing – true 3D bioprinting. As such, we focused on the methods most commonly used for vascular restorations with an internal lumen diameter in the range of 1 mm to 6 mm.

  1. Quoted more outdated references, please refer to more novel references.
  • In the actual description of the methods used for the production of vascular substitutes, all the articles that could be traced and met our criteria for inclusion in the summary (i.e. production of tubular vascular structures with a diameter of at least 1 mm, possible as implantable construct in vivo) are mentioned, while the first attempts date back to the turn of the millennium and are significant milestones in this field, which is why we found it important to mention them in the summary. In the actual text section for the description of each issue, we try to include recent articles from about 2015 onwards, as the field has undergone a rapid evolution over the last 10 years. We have therefore revised all the references and replaced older references with new ones.

  1. It is mentioned in the article that different methods can be used to improve the performance of the prepared samples, but the article does not reflect how the performance changes. For example, lines 209-214 mention that physical or chemical crosslinking can be used to compensate for the lack of mechanical strength, and can increase the robustness and resistance of the graft. There is no exact data to reflect the performance changes, please compare the performance data.
  • The paragraph about chemical and physical crosslinking was added in chapter 3. Next, in the description of collagen crosslinking, a reference to a study comparing the mechanical properties of crosslinked collagen with native collagen has been added.

  1. Are there sufficient discussions on the future development of polymer hydrogel bioprinting to prepare vascular grafts in the literature? Or does it rely on speculative statements that are not supported by sufficient research?
  • There are several studies in the literature that deals exclusively with the future of artificial vascular replacement development in tissue engineering: 3389/fphys.2022.1079421, 10.1089/ten.teb.2015.0100, 10.1161/CIRCRESAHA.121.319984.

In general, they reflect on the following challenges in this field:

  • availability of materials and technologies, economic viability,
  • biocompatibility, immunomodulation, and graft remodeling in tissue,
  • and the synergistic effects of grafts, cells, mechanical, and immunological reactions.

This issue is very broad, so we have tried to outline the most important points under discussion, which we have quoted from the source. Our assumptions, which also appear in the discussion, are based on our own work in the field of vascular replacement.

Reviewer 3 Report

Comments and Suggestions for Authors

The manuscript is titled ‘Current status of bioprinting using polymer hydrogels for the  production of vascular grafts.’ This review article offers a comprehensive summary of the current state of bioprinting for the production of vascular grafts. It also summarizes the hydrogels that are employed as bioinks, the materials of the carriers, and the current methods of fabrication, particularly for vessels that are smaller than 6 mm, which are the most difficult to replace in cardiovascular systems. However, it is essential to offer further clarification on specific issues. Consequently, it is advised that this paper be published after minor revisions.

More research should be done regarding the type of polymers used as hydrogels in bioprinting, there are other types of polymers such as KERMA, cellulose, etc. which were not mentioned in the review, and should definitely be included.

A few bioprinting techniques were explained in the review. Other techniques were mentioned shortly, such as laser-assisted bioprinting. It would be better if a few of the latest references using this method were given as an example, and if advantages and disadvantages were mentioned.

 A table of bioprinting methods and a table of polymers used in bioprinting technology that summarizes overall strengths, weaknesses, pros. and cons. of each along with references should make the review clearer.

Line 120 à The title “2.3. D bioprinting for vascular replacements” seems to have a mistake in numbering. Also, do you mean 3D printing or D printing?

Author Response

The manuscript is titled ‘Current status of bioprinting using polymer hydrogels for the  production of vascular grafts.’ This review article offers a comprehensive summary of the current state of bioprinting for the production of vascular grafts. It also summarizes the hydrogels that are employed as bioinks, the materials of the carriers, and the current methods of fabrication, particularly for vessels that are smaller than 6 mm, which are the most difficult to replace in cardiovascular systems. However, it is essential to offer further clarification on specific issues. Consequently, it is advised that this paper be published after minor revisions.

 Thank you for your appreciation of our article and the valuable comments we have been inspired by.

More research should be done regarding the type of polymers used as hydrogels in bioprinting, there are other types of polymers such as KERMA, cellulose, etc. which were not mentioned in the review, and should definitely be included.

  • In this article, we focused on currently used methods for the production of vascular grafts using 3D bioprinting. The criteria are very narrow, as the graft produced must be tubular in shape with a diameter of at least 1 mm, and at the same time the cells and hydrogels must be applied using the bioprinting technique. This has led to the exclusion of many articles dealing with the production of substitutes, including substrates made up of cellulose or KERMA. Therefore, they were not even mentioned in the general section.

A few bioprinting techniques were explained in the review. Other techniques were mentioned shortly, such as laser-assisted bioprinting. It would be better if a few of the latest references using this method were given as an example, and if advantages and disadvantages were mentioned.

  • These methods are mentioned in section 4.1.3, where the principle of the method and the reasons why the method is not widely used for bioprinting vascular substitutes are presented. We found only one application of this method that matched our selection criteria, so we did not describe the method further, as the article is intended to focus mainly on the application of bioprinting for vascular replacements and a brief description of the methods and materials used for this purpose.

 A table of bioprinting methods and a table of polymers used in bioprinting technology that summarizes overall strengths, weaknesses, pros. and cons. of each along with references should make the review clearer.

  • Thank you for this comment as it inspired us to create a table that includes applications for bioprinting in tissue vascular replacements. We did not directly create a table of methods, materials, and their advantages and disadvantages, as our article is not directly focused in this direction and, moreover, many articles on this topic can be found.

Line 120 à The title “2.3. D bioprinting for vascular replacements” seems to have a mistake in numbering. Also, do you mean 3D printing or D printing?

  • Numbering was corrected.
